# Oral intake of mesoporous silica is safe and well tolerated in male humans

Emilia Hagman[1], Amira Elimam[1], Natalia Kupferschmidt[2], Kerstin Ekbom[1], Stephan Rössner[3,4], Muhammad Naeem Iqbal[4], Eric Johnston[4], Maria Lindgren[4], Tore Bengtsson[2,4], Pernilla Danielsson[1]*

1 Department of Clinical Science, Intervention and Technology, Karolinska Institutet, Stockholm, Sweden, 2 Department of Molecular Biosciences, Stockholm University, Stockholm, Sweden, 3 Apple Bay Obesity Research Centre, Bromma, Sweden, 4 Sigrid Therapeutics AB, Stockholm, Sweden

* pernilla.danielsson.liljeqvist@ki.se

## Abstract

### Background

Precisely engineered mesoporous silica has been shown to induce weight loss in mice, but whether it is safe to use in humans have not investigated.

### Objective

The aim was to determine whether oral dosing, up to 9 grams/day, of precisely engineered mesoporous silica as a food additive can be used safely in male humans.

### Design

This single blinded safety study consisted of two study arms including 10 males each (18–35 years). One arm consisted of participants with normal weight and one with obesity. After a placebo run-in period, all subjects were given porous silica three times daily, with increasing dose up to 9 grams/day (Phase 1). Subjects with obesity continued the study with highest dose for additional 10 weeks (Phase 2).

### Results

All participants completed Phase 1 and 90% completed Phase 2, with approximately 1% missed doses. Participants reported no abdominal discomfort, and changes in bowel habits were minor and inconsistent. The side effects observed were mild and tolerable, biomarkers did not give any safety concern, and no severe adverse events occurred.

### Conclusion

Mesoporous silica intake of up to 9 grams/day can be consumed by males without any major adverse events or safety concerns.

**Data Availability Statement:** All relevant data are within the manuscript and its Supporting Information files.

**Funding:** This study was supported by Sigrid Therapeutics AB https://www.sigridthx.com/. The company provided the compound studied and sponsored Division of pediatrics, CLINTEC, Karolinska Institutet to conduct the study and external companies for analyzing urine. Sigrid Therapeutics AB was partly involved in the study design and manuscript preparation but were not involved in data collection, data analysis or decision to publish. Sigrid Therapeutics AB also provided some support in the form of salaries or other form of remuneration at the time of the study for authors NK, SR, EJ, ML, MNI and TB. The specific roles of these authors are articulated in the 'author contributions' section.

**Competing interests:** The following authors have the following competing interests: NK, SR, EJ, ML, MNI and TB are or have been connected to Sigrid Therapeutics AB (employed, consultant or advisory board). This commercial affiliation does not alter our adherence to PLOS ONE policies on sharing data material. EH, AE, KE, and PD declared that no competing interests exist.

## Introduction

The first-line treatment for obesity includes lifestyle modification targeting diet and physical activity, however, the effects observed have been modest [1], which is why bariatric surgery treatment has increased considerably over the last decades. To cover the gap between lifestyle modification and bariatric surgery, new ways to treat obesity are needed [2]. The pharmacological treatment of obesity has faced several problems, such as having potential severe side effects [3, 4] causing retraction of such compounds from the market, and uncertain effects by the few new compounds available today [5]. Safe and well tolerable alternatives are needed.

The material tested here as a food additive may be one way to enhance the effect of lifestyle modifications, both for treatment of obesity and for obesity prevention in overweight participants. Another closely related indication, where treatments with few or no adverse events are needed, is for patients with type-2 diabetes. Mesoporous silica compounds with narrow pore size distributions have been shown to induce weight loss in mice [6]. Unlike commercial food grade silica (E551), these compounds have narrow pore size distributions, and are therefore referred to as precisely engineered porous silica. In addition, diatomaceous earth (composed of amorphous silicates from sedimentary rock) has been shown to lower blood lipids in subjects with hypercholesterolemia in an open uncontrolled eight weeks study [7]. The mechanisms of how porous silica exerts its effects, inducing weight loss and lipid lowering are still to be investigated. However, possible mechanisms include that the porosity and large surface area of porous silica materials facilitates absorption of biomolecules into the porous material. This absorption is probably a combination of specificity dependent on pore size and unspecific interactions due to the material's large surface area, and physicochemical characteristics such as charge and porosity. Gastrointestinal enzymes such as lipases have been shown to be specifically absorbed in silica pores of well-defined size [8]. One could hypothesis that such an absorption will reduce the enzymatic activity and subsequently reduce the gastrointestinal uptake of nutrients in vivo.

Based on the literature on orlistat side effects, the expected side effects of porous silica may be diarrhea due to reduced gastrointestinal lipase activity. In addition, lower gut absorption of vitamins and trace elements leading to deficiencies may theoretically be expected [9]. Silica compounds have been widely applied as excipients in dietary supplements, pharmaceutical products and cosmetics. Synthetic amorphous silica is described in the U.S. Pharmacopeia and approved as food additives (E551) under EU regulations. Their use is "generally recognized as safe" (GRAS) by the Food and Drug Administration, USA. The present limit that may be safely used in food under 21 CFR 172.480 is 2% by weight of the food. Hence, silica intake has been defined as safe. However, precisely engineered porous silica differs from other food grade silica (E551) evaluated to date by its narrower pore size distribution and it has been described to induce weight loss and lower fat content in mice [6] through absorption of biomolecules into the porous material. Therefore, it is possible that the precise pore size has an effect on the safety profile of this engineered material and in particular, this material might affect gastrointestinal functions.

The aim of the present study was thus to determine whether oral dosing, up to 9 grams per day, of specifically designed porous silica compounds can be safely used in normal weight and obese male humans, without significant side effects on gastrointestinal function, bowel emptying habits, and biomarkers.

## Materials and methods

### Participants

This single blinded uncontrolled First-In-Man study with a placebo run-in period consisted of two study arms including 10 young males each, between 18 and 35 years of age, flowchart

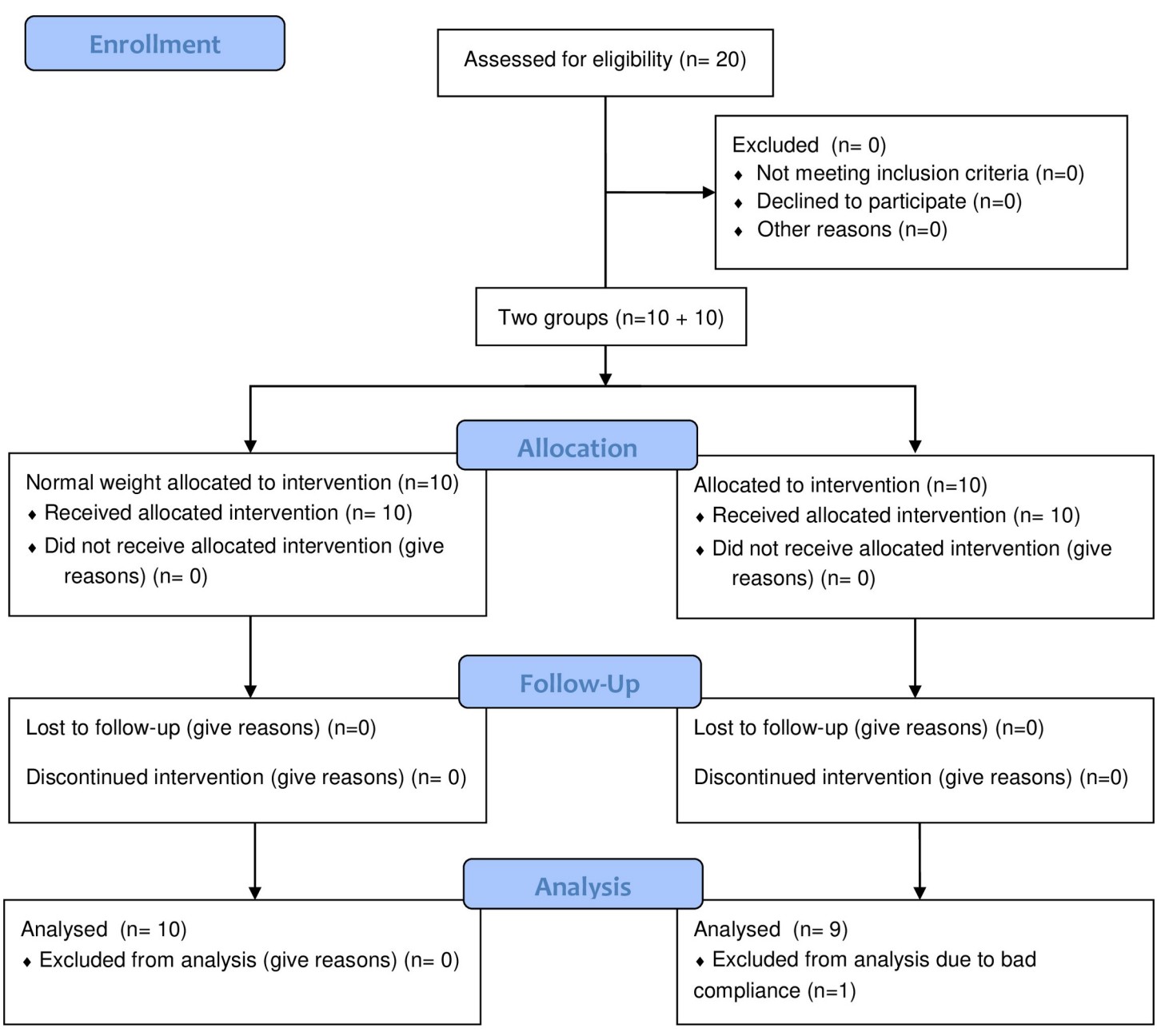

**Fig 1. Participant flowchart.**

Fig 1. One arm consisted of normal weight participants with BMI ranges between 20.0–25.0 kg/m$^2$ and one arm with participants with obesity, BMI ranges between 30.0–45.0 kg/m$^2$. All included subjects were recruited via advertisement, and the study was performed August 17th to December 21st 2015 in Stockholm, Sweden. Exclusion criteria for both study arms included; chronic somatic diseases that may affect metabolic and/or gastro-intestinal function (e.g. diabetes, hypertension, dyslipidemia, inflammatory bowel disease, gluten intolerance, pancreatic dysfunction, other causes of malabsorption, neoplastic disease), allergies with previous anaphylactic reactions, previous abdominal surgery, and current or previous history of eating disorders. Further exclusion criteria include; restrictive diets (e.g. very low carbohydrate or

vegan) during the past year, psychiatric disorders that may influence adherence (e.g. schizo-phrenia), drug or alcohol abuse, continuous pharmacological treatment that might influence the study outcome, and other conditions which the investigator considered could negatively affect the outcome of the study or study adherence. The study was approved by the Regional Ethical Review board, Stockholm, Sweden in May 21, 2015, file no. 2015/593-31. The methods were carried out in accordance with this approval. All research was performed in accordance with the recommendations of the Declaration of Helsinki. The study has also been registered in Clinical Trials Registry (clinicaltrials.gov, ID: NCT03667430, Date: 11/09/2018). The recommendations regarding whether phase 1 studies should be registered or not is not uniform, why the registration of the current trial was done after study initiation. Initially, we chose to comply with the European regulations (https://www.eortc.be/services/doc/clinical-eu-directive-04-april-01.pdf). However, later Sigrid Therapeutics AB requested the study to be registered as suggested by some other international recommendations. Further, Sigrid Therapeutics AB confirms that all ongoing and related trials for this compound are registered.

## Test-items

Precisely engineered mesoporous silica compounds were synthesized by a modified method as reported previously [10]. In brief, P123 triblock copolymer (with average molecular weight = 5800 g/mol, $EO_{20}PO_{70}EO_{20}$) a mesostructured templating agent was dissolved in aqueous hydrochloric acid (HCl). Complete dissolution of P123 was followed by the addition the silica source tetraethyl orthosilicate (TEOS) under vigorous stirring at 40˚C. Final molar ratio of the solution was P123: TEOS: HCl: $H_2O$: 0.02: 1.00: 6.32: 234.62. The synthesis was kept static at 40˚C for 20 hours followed by hydrothermal treatment for 17–50 hours at 100˚C. Filtered, washed and dried material was subjected to calcination (550˚C in air) to remove the organic template and generate the open porous network. Nitrogen sorption analysis was performed in order to characterize the silica particles in terms of surface area, pore volume and pore size. The analysis was performed at liquid nitrogen temperature (-196˚C) using a TriStar II volumetric adsorption analyser (Micromeritics Instrument Corp., GA, USA). Brunauer–Emmett–Teller (BET) surface area was calculated from adsorption isotherm at a relative pressure (p/p˚) of <0.2. Total pore volume was recorded at a relative pressure (p/p˚) = 0.99. Pore size was obtained by applying the density functional theory (DFT) method assuming a cylindrical pore model. Pore structure was characterized by low-angle X-Ray diffraction (XRD) on a powder PANalytical diffractometer (PANalytical, Karlsruhe, Germany) operated at 45Kv and 40mA, with 0.02˚ step size and equipped with Cu Kα radiation source. Scanning electron microscopy (SEM) using a JEOL JSM-7401F (JEOL Ltd., Tokyo, Japan) was used to characterize the particle agglomerates and morphology. The silica was delivered to participants as powder in vials containing 1.0–3.0 grams of the silica per portion to be mixed with water in each powder containing vial. The participants were instructed to drink a large glass (approximately 250 milliliter) of water with the powder. Cellulose powder (VIVAPUR® MCC Microcrystalline cellulose) was used as placebo and provided in identical looking placebo vials. The placebo was given blinded i.e. single blinded in that the healthy volunteers were not informed about the placebo run-in.

## Material characterisation

Particle morphology determined by SEM revealed large agglomerates of several micrometers composed of rod-shaped particles approximately 1–3 x 0.4–0.5 micrometers (Fig 2A). The BET surface area of the silica particles was in the range 600–884 $m^2$/g and total pore volumes ranged from 0.6 to 1.1 $cm^3$/g. The nitrogen adsorption-desorption isotherm and pore size

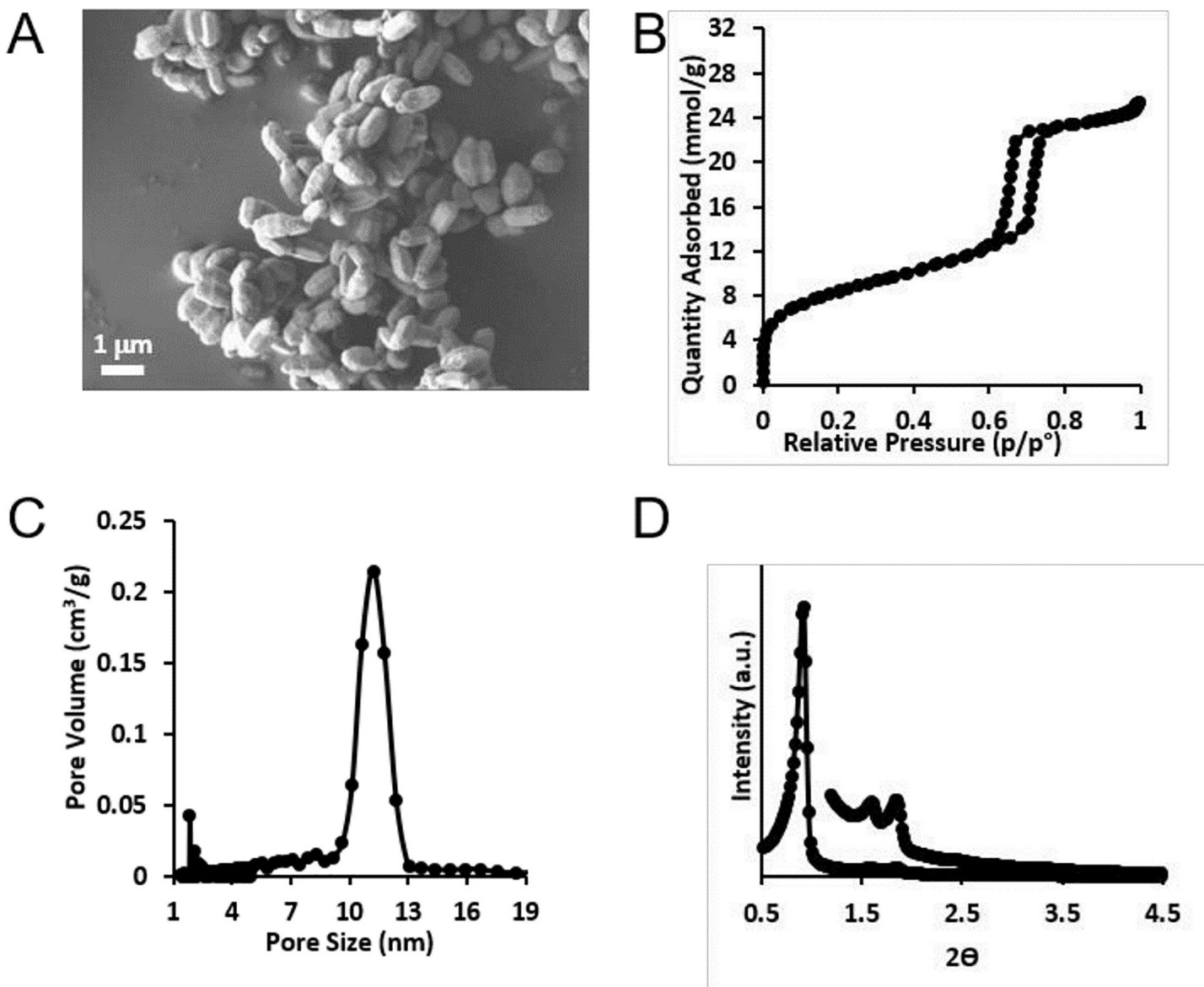

**Fig 2.** A-D. SEM micrograph (A), nitrogen adsorption-desorption isotherms (B), pore size distribution measured via density functional theory (DFT) (C) and low angle XRD patterns with peaks 110, and 200 raised by a factor of 5 for clarity purposes (D) of a representative batch of the mesoporous silica used in this study.

distribution data of a representative batch are presented in Fig 2B and 2C. The mean pore size of the studied silica was in the range 7–12 nm. The low angle XRD pattern in Fig 2D show peaks that can be indexed on the basis of 2D hexagonal pore geometry in accordance with previous reports for this class of silica mesoporous particles [11]. The compound had high purity and conforms to the test requirements as published to date by the U.S. Pharmacopeia for silicon dioxide and meets food additive standards for E551 under EU Regulation No. 231/2012.

## Procedure

All participants received written and oral information about the study. After signing informed consent, a medical examination (Table 1) was performed. Information regarding eating habits,

**Table 1. Description of study timeline and procedures performed.**

| Phase 1 | | | | | Phase 2 | | | |
|---|---|---|---|---|---|---|---|---|
| | Baseline | Week 1 | Week 2 | Week 3 | | Week 4 | Week 8 | Week 12 |
| | Day 1 | Day 7 | Day 14 | Day 21 | | | | |
| Day | 1–5 | 6–9 | 10–14 | 15–21 | | | | |
| | Placebo | Silica 1 g | Silica 2 g | Silica 3 g | | Silica 3 g | Silica 3 g | Silica 3 g |
| Weight | x | x | x | x | Weight | x | x | x |
| Clinical control | x | x | x | x | Clinical Control | x | x | x |
| Blood Sample | x | x | x | x | Blood Sample | x | x | x |
| Urine Sample | x | | | x | Urine Sample | x | x | x |
| Feces Sample | x | | | x | Feces Sample | x | | x |
| Adverse events | | x | x | x | Adverse events | x | x | x |

sleep patterns, living conditions and gastrointestinal health were obtained from written forms and orally.

## Phase 1, 21 days

Both study arms received placebo in the same vials study day 1–5 (five days run-in period). Thereafter all participants received increasing doses of porous silica as follows; Day 6–9, 1 gram, three times daily, day 10–14, 2 grams, three times daily, and day 15–21, 3 grams, three times daily. The maximum dose of 9 grams per day was selected based on the lowest estimated dose recognized as safe according to Food and Drug Administration (FDA); 2% (10–30 grams) of daily food intake (500–1500 grams) [12].

Clinical examinations and blood sampling were performed day 1, 7, 14, and 21. Faeces and morning urinary samples were collected on day 1, and 21. If gastrointestinal adverse events would occur after an increase in dosage, the study staff had a routine follow-up to adapt the dosage protocol to facilitate adherence.

## Phase 2, 10 additional weeks

After Phase 1, completers from the arm with participants with obesity continued with the individually highest tolerated dose tested in Phase 1. This phase continued for 10 additional weeks, i.e., in total 12 weeks with the maximum tolerated dose (three grams of porous silica, three times daily). During phase 2, participants were examined by the following protocol: clinical examinations, blood sampling, morning urinary sampling week 4, 8, 13, and faeces sampling week 4, 13. Throughout phase 2 the trial staff maintained and documented weekly contact with the participants by phone for potential dose adjustments, dietary advice and documentation of adverse events.

## Clinical examinations

An experienced registered nurse performed the measurements of weight to the nearest 0.1 kg (Tanita BC-418, Tanita Corp.; Tokyo Japan), height to the nearest 0.1 cm (SECA model 264, Seca, Hamburg, Germany), and blood pressure manually. A medical doctor performed cardio-respiratory and abdominal examinations. At baseline a food frequency questionnaire (FFQ) was obtained. At all visits questions regarding adherence, life style changes, gastrointestinal function/habits and adverse events were asked and answered.

Fasting blood samples included in whole blood; leukocyte concentration, MCH, platelets, mean cell [erythrocytes] volume (MCV), erythrocyte concentration, erythrocyte volume

fraction (EVF), hemoglobin, in plasma; C-reactive protein (CRP), aspartate aminotransferase (AST) alanine aminotransferase (ALT), calcium, magnesium, creatinine, in serum, in serum; retinol, 25-OH vitamin D and zinc. Faeces analyses included hemoglobin, calprotectin and elastase lipids. All above clinical examinations and tests were performed and run at Karolinska University Hospital and Karolinska University Hospital Laboratory, Stockholm, Sweden. Silica concentration, measured as silicon content, in urine was measured with ICP-SFMS by ALS Scandinavia AB, Luleå, Sweden.

## Adherence & adverse events

The participants brought all empty vials back to the clinic in order to ensure adherence. A summary table of incidence and observed number of adverse events is presented by part and by group and sorted by system organ class (SOC) and by preferred term (PT). MedDRA V19.1 was used for coding.

## Statistics

Descriptive statistics are presented with mean-, min- and max values. To investigate differences in anthropometrical measures and biomarkers from baseline to follow-up visits, paired t-tests were used to assess differences in anthropometrical measures and biomarkers from baseline to follow-up visits. Only reported values were used for the purpose of these analyses, i.e. no data were imputed. All analyses were performed in STATISTICA (Statsoft Inc., Tulsa, USA).

## Results

### Descriptive and adherence

In the arm with normal weight participants, the average age was 26.2 years and the average BMI was 23.1 kg/m$^2$. The corresponding numbers in the arm consisting of obese participants were 28.3 years and 34.9 kg/m$^2$. In both arms all baseline biomarkers were within the normal ranges. The laboratory data confirmed that all participants were metabolically healthy, despite their obesity. The complete baseline profiles of the two arms are presented in Table 2. All participants in both arms completed Phase 1, and 9 out of 10 participants in arm with obese participants completed Phase 2. One participant had insufficient adherence during Phase 2 and was therefore excluded from the week 10 analysis. All participants could follow the dose increase program without problems, hence no dose adjustments from predefined treatment schedule were made. The adherence to take the silica was generally good. Approximately 1% of the porous silica vials were missed during Phase 1 and Phase 2, Table 3.

### Anthropometrics

In the arm with normal weight participants no changes were noted as regards to body weight BMI, or blood pressure. Likewise, in the arm with participants with obesity no changes were observed during Phase 1 or Phase 2, except for a decrease in diastolic blood pressure from baseline to week 12. The complete follow-up profiles of the two arms are presented in Table 2. All participants were interviewed about potential changes in eating patterns and physical activity throughout the study, but no consistent pattern of change was observed. Likewise, questions were asked about changes in sleeping patterns, but no systematic pattern of change was reported.

                                                                                                                              

**Table 2. Data of anthropometry, blood pressure, biomarkers, urine and faeces samples and kidney function at baseline, end of Phase 1 (Day 21) and end of Phase 2 (Week 12).**

| | Normal weight | | | | | Obese | | | | | | | |
| --- | --- | --- | --- | --- | --- | --- | --- | --- | --- | --- | --- | --- | --- |
| | Baseline | | Day 21 | | | Baseline | | Day 21 | | | Week 12 | | |
| | n | Mean (SD, Min-Max) | n | Mean (SD, Min-Max) | p value | n | Mean (SD, Min-Max) | n | Mean (SD, Min-Max) | p value | n | Mean (SD, Min-Max) | p value |
| *Anthropometric* | | | | | | | | | | | | | |
| Age (year) | 10 | 26.2 (3.5, 23–31) | 10 | | | 10 | 28.3 (4.6, 21–33) | | | | | | |
| Weight (kg) | 10 | 78.7 (4.7, 71.6–85.0) | 10 | 78.8 (5.0, 69.8–84.4) | 0.75 | 10 | 118.7 (11.4, 99.8–139.1) | 10 | 119.1 (11.0, 101.3–138.5) | 0.44 | 9 | 117.9 (11.1, 102.2–138.8) | 0.59 |
| BMI | 10 | 23.1 (1.7, 19.8–25.0) | 10 | 23.1 (1.8, 19.9–25.1) | 0.85 | 10 | 34.9 (2.5, 31.5–38.5) | 10 | 35.0 (2.5, 31.5–38.7) | 0.30 | 9 | 34.9 (2.8, 31.9–39.7) | 0.53 |
| *Blood Pressure* | | | | | | | | | | | | | |
| Systolic (mm Hg) | 10 | 118.5 (8.2, 110–130) | 10 | 118.3 (4.1, 110–125) | 0.94 | 10 | 119 (6.6, 110–125) | 10 | 123.5 (7.8, 110–135) | 0.18 | 9 | 123.3 (7.1, 120–140) | 0.28 |
| Diastolic (mm Hg) | 10 | 73 (5.9, 60–80) | 10 | 68.5 (7.5, 60–80) | 0.12 | 10 | 80 (4.6, 70–85) | 10 | 74.5 (8.0, 60–80) | 0.16 | 9 | 73.3 (6.1, 60–80) | 0.007 |
| *Biomarkers* | | | | | | | | | | | | | |
| Leukocytes (x10(9)/L) | 9 | 5.9 (0.9, 4.8–7.8) | 10 | 5.9 (1.2, 3.9–7.6) | 0.60 | 10 | 6.8 (1.3, 5.2–8.8) | 10 | 6.7 (1.3, 4.5–8.6) | 0.80 | 9 | 7.0 (1.7, 5.2–10.4) | 0.92 |
| Erytrocytes (x10(12)/L) | 10 | 5.3 (0.4, 4.7–5.8) | 10 | 5.0 (0.4, 4.2–5.6) | 0.004 | 10 | 5.3 (0.3, 4.9–5.9) | 10 | 5.2 (0.2, 4.7–5.6) | 0.009 | 9 | 5.2 (0.2, 4.9–5.4) | 0.07 |
| Hb (g/L) | 10 | 154.2 (11.6, 137.0–174.0) | 10 | 145.4 (8.1, 128.0–156.0) | 0.010 | 10 | 154.3 (9.1, 140.0–166.0) | 10 | 150.7 (10.2, 133.0–166.0) | 0.016 | 9 | 150.5 (9.1, 139.0–171.0) | 0.11 |
| EVF | 10 | 0.46 (0.03, 0.42–0.51) | 10 | 0.44 (0.02, 0.40–0.47) | 0.003 | 10 | 0.46 (0.03, 0.41–0.50) | 10 | 0.45 (0.03, 0.40–0.47) | 0.005 | 9 | 0.44 (0.02, 0.41–0.49) | 0.04 |
| Erc(B)-MCV (fL) | 10 | 87.8 (3.5, 83.0–95.0) | 10 | 87.6 (2.7, 84.0–93.0) | 0.62 | 10 | 86.2 (4.0, 79.0–93.0) | 10 | 85.8 (3.4, 79.0–91.0) | 0.27 | 9 | 85.4 (3.6, 79.0–91.0) | 0.28 |
| Erc(B)-MCH (pg) | 10 | 29.1 (1.0, 28.0–31.0) | 10 | 29.4 (1.1, 28.0–31.0) | 0.28 | 10 | 28.8 (1.7, 26.0–32.0) | 10 | 29.1 (1.5, 27.0–32.0) | 0.08 | 9 | 28.9 (1.7, 26.0–32.0) | 0.35 |
| Platelets (x10(9)/L) | 10 | 231.8 (34.7, 177.0–301.0) | 10 | 235.5 (41.8, 175.0–321.0) | 0.35 | 10 | 283.8 (57.1, 189.0–376.0) | 10 | 272.8 (48.3, 190.0–337.0) | 0.17 | 9 | 264.6 (54.3, 176.0–349.0) | 0.0501 |
| Calcium (mmol/L) | 9 | 2.26 (0.08, 2.11–2.34) | 10 | 2.21 (0.10, 2.06–2.37) | 0.13 | 10 | 2.28 (0.06, 2.19–2.40) | 10 | 2.23 (0.09, 2.10–2.34) | 0.046 | 9 | 2.28 (0.12, 2.05–2.40) | 0.90 |
| AST (microkat/L) | 10 | 0.58 (0.20, 0.41–1.12) | 10 | 0.59 (0.54, 0.34–2.08) | 0.95 | 10 | 0.38 (0.08, 0.27–0.51) | 10 | 0.48 (0.29, 0.27–1.26) | 0.23 | 9 | 0.44 (0.14, 0.29–0.78) | 0.11 |
| ALT (microkat/L) | 10 | 0.39 (0.08, 0.24–0.52) | 10 | 0.37 (0.09, 0.30–0.60) | 0.65 | 10 | 0.43 (0.11, 0.20–0.60) | 10 | 0.61 (0.46, 0.24–1.69) | 0.21 | 9 | 0.46 (0.16, 0.25–0.83) | 0.54 |
| Magnesium (mmol/L) | 10 | 0.85 (0.08, 0.68–0.92) | 10 | 0.83 (0.06, 0.71–0.90) | 0.33 | 10 | 0.85 (0.03, 0.81–0.91) | 10 | 0.85 (0.04, 0.79–0.91) | 0.70 | 9 | 0.79 (0.14, 0.42–0.89) | 0.21 |
| CRP (mg/L) | 10 | 2.3 (2.8, 0.2–8.4) | 10 | 0.4 (0.3, 0.2–1.1) | 0.09 | 10 | 4.1 (5.1, 0.5–16.7) | 10 | 3.8 (5.6, 0.3–18.9) | 0.63 | 9 | 21.3 (54.3, 1.0–166) | 0.34 |
| Zinc (micromol/L) | 7 | 12.7 (4.0, 4.8–17) | 10 | 13.1 (1.3, 11.0–15.0) | 0.37 | 9 | 16.4 (3.2, 11.0–21.0) | 10 | 12.8 (1.3, 11.0–15.0) | 0.002 | 9 | 13.8 (1.1, 12.0–15.0) | 0.018 |
| Vitamin D (mmol/L) | 10 | 71.2 (17.9, 37.0–90.0) | 10 | 59.5 (14.9, 36.0–78.0) | 0.007 | 10 | 49.7 (17.3, 17.0–73.0) | 10 | 45.8 (13.8, 18.0–62.0) | 0.12 | 9 | 46.0 (6.4, 38.0–57.0) | 0.33 |
| Retinol (micromol/L) | 10 | 1.8 (0.3, 1.2–2.1) | 10 | 1.6 (0.3, 1.2–2.0) | 0.024 | 10 | 1.6 (0.2, 1.2–2.0) | 10 | 1.7 (0.5, 1.2–2.8) | 0.32 | 9 | 1.7 (0.4, 0.9–2.5) | 0.61 |
| *Urine* | | | | | | | | | | | | | |
| Silica (mg/L) | 10 | 11.3 (6.4, 4.7–21.1) | 10 | 60.1 (25.8, 25.2–98.5) | < 0.001 | 10 | 14.8 (12.9, 3.8–44) | 10 | 79.8 (44.8, 11.4–148.3) | < 0.001 | 9 | 48.9 (40.3, 16.8–134.7) | 0.051 |
| *Faeces* | | | | | | | | | | | | | |
| Calprotectin (mg/kg) | 8 | 17.6 (13.3, 6.0–40.0) | 10 | 10.2 (5.6, 5.0–20.0) | 0.1007 | 9 | 38.4 (45.7, 5.0–148.0) | 10 | 21.1 (23.2, 5.0–81.0) | 0.08 | 9 | 25.4 (22.0, 5.0–61.0) | 0.45 |
| Elastase (microg/g) | 10 | 497.4 (8.2, 474.0–500.0) | 10 | 500.0 (0.0, 500.0–500.0) | 0.34 | 9 | 499.4 (1.7, 495.0–500.0) | 10 | 470.8 (67.9, 293.0–500.0) | 0.21 | 9 | 478.3 (33.0, 409.0–500.0) | 0.09 |

*(Continued)*

                                                          

**Table 2.** (Continued)

| | n | Normal weight Baseline Mean (SD, Min-Max) | n | Day 21 Mean (SD, Min-Max) | p value | n | Obese Baseline Mean (SD, Min-Max) | n | Day 21 Mean (SD, Min-Max) | p value | n | Week 12 Mean (SD, Min-Max) | p value |
|---|---|---|---|---|---|---|---|---|---|---|---|---|---|
| Hb | 10 | Negative | 10 | Negative | | 10 | Negative | 10 | Negative | | 9 | Negative | |
| *Kidney function* | | | | | | | | | | | | | |
| Creatinine (micromol/L) | 10 | 84.8 (6.4, 77.0–93.0) | 10 | 83.2 (6.7, 71–93) | 0.37 | 10 | 84.0 (16.5, 69.0–118.0) | 10 | 84.7 (11.5, 64.0–103.0) | 0.82 | 9 | 76.0 (10.1, 62.0–92.0) | 0.025 |
| Cystatin C (mg/L) | 10 | 0.84 (0.07, 0.76–0.96) | 10 | 0.78 (0.07, 0.69–0.89) | 0.0007 | 10 | 0.88 (0.12, 0.71–1.08) | 10 | 0.87 (0.12, 0.69–1.06) | 0.19 | 9 | 0.91 (0.18, 0.70–1.31) | 0.631 |
| eGFR CysC /(mL/min/1.7)) | 10 | 89.7 (0.9, 87.0–90.0) | 10 | 90.0 (0.0, 90.0–90.0) | 0.34 | 10 | 86.8 (6.0, 73.0–90.0) | 10 | 87.0 (5.4, 74.0–90.0) | 0.72 | 9 | 85.0 (11.2, 57.0–90.0) | 0.567 |

SD = standard deviation.

## Biomarkers

**Arm with normal weight participants.** In the arm with normal weight participants, a minor decrease, yet within normal range, of erythrocytes (p = 0.004) and EVF (p = 0.003) was observed. Further, a 6% drop in hemoglobin concentration (p = 0.009) occurred, resulting in that one participant ended up with a value under the normal range for sex and age (128 g/L). Both retinol (p = 0.024) and 25-OH vitamin D (p = 0.007) decreased on average during Phase 1, resulting in four participants with vitamin D levels below normal range (75 mmol/L). All participants stayed within the normal range of retinol levels (1.0–3.3 µmol/L). All other bio-markers remained unchanged within normal levels. In summary, although erythrocytes, hemoglobin and EVF showed statistically significant reductions during Phase 1, the changes observed are within normal range and show no signal for safety concerns.

**Arm with participants with obesity.** In the arm with obese participants the changes in the erythrocyte profile (erythrocytes, EVF and hemoglobin) were similar to the participants in the normal weight arm during Phase 1. However, the erythrocytes and hemoglobin concentration returned to initial values during the 10 weeks of Phase 2. Even though the EVF remained decreased after Phase 2, all participants stayed within the normal range (0.39–0.50). Average zinc levels decreased among the obese participants during Phase 1 from 16.4 to 12.8 µmol/L (p = 0.002) and remained decreased, but within the normal range, until the end of Phase 2 (13.8 µmol/L, p = 0.018). In contrast to the normal weight participants, the participants with obesity did not change their retinol and 25-OH vitamin D levels.

## Kidney function

The creatinine levels were unchanged in the arm with normal weight participants (p = 0.37). Among the obese participants, the creatinine levels decreased from baseline (84.0 µmol/L) to

**Table 3.** Adherence of silica intake.

| | Period | Treatment | Number of failed vials (number of participants) | Total number of planned vials |
|---|---|---|---|---|
| Phase 1 (n = 20) | Week 1 or Day 1–5 | Placebo | 4 (4) | 360 |
| | Week 2 or Day 6–9 | Silica 1 g | 6 (3) | 450 |
| | Week 3 or Day 10–21 | Silica 2–3 g | 5 (4) | 450 |
| Phase 2 | Week 4–7 | Silica 3 g | 6 (1) | 900 |
| (n = 9) | Week 8–12 | Silica 3 g | 16.5 (4) | 900 |

Phase 2 follow-up (76.0 μmol/L), (p = 0.025), resulting in a normalization of all participants' creatinine values. On an average level, cystatin C and eGFR remain unchanged for both arms. However, one participant with obesity had a slightly elevated creatinine level at baseline (106 μmol/L) and increased his cystatin C level to an abnormal level for his age (1.31 mg/L) by the end of Phase 2. Calculated eGRF based on different methods at the end of Phase 2 showed however large differences; eGFR based on cystatin C was 57 ml/min/1.73 m$^2$ whereas eGFR based on creatinine was 87.4. This participant was followed up with further investigations of his kidneys (iohexol clearance test) at five months post the end of Phase 2, and the kidney function showed normal activity. The baseline values might indicate that impaired or fluctuating kidney function could already have been present at study initiation.

## Bowel function

The average of stool/defecation frequency was two times per day at baseline. The range in bowel emptying frequency was between 0.6–6 times per day. In both arms the majority of the participants reported unchanged frequency of defecation. Some participants reported more and others less frequent defecation patterns. No participants reported feelings of pain or nausea. Reports of inflated stomach were sporadic and were not more frequently reported with increased dose of porous silica. In this limited study, the majority of both normal weight (6/10) and obese (6/10) on average reported no changes in stool consistency. The presence of hemoglobin in the feces was determined before and during the study (baseline, end of Phase 1 and end of Phase 2). No samples tested positive for hemoglobin.

## Silica in urine

Paired t-tests were used to compare baseline data with day 21 for silica in urine. In both groups the levels of silica in urine from baseline to day 21 was increased (p≤0.001 for both arms). The urine level was highly variable, and the level after Phase 2 in the arm with obese participants reached slightly less than five times the baseline value (p = 0.051).

## Adverse events

Normal weight participants reported unspecific minor health problems, which did not seem to be related to the intake of porous silica and some general gastro-intestinal discomfort. Of the obese participants, some also reported unspecific problems and a variety of gastro-intestinal symptoms. One participant complained of sensing the smell and taste of silica during the first week of Phase 2. This later disappeared, see Table 4.

# Discussion

In this safety study, we show that an oral intake of up to 9 grams per day of porous silica can be consumed without any major adverse events or safety concerns. None of the study participants reported any changes in diet, physical activity or sleep patterns during the study period as reported in the follow-up questions at each visit. Since one theory is that mesoporous silica interacts with digestion enzymes such as lipase, several questions were asked about the gastro-intestinal habits, since the side effects of lipase inhibiting agents may affect the gastrointestinal system. The results from this study showed mixed results with both harder and looser stools. However, in general, these side effects were mild and tolerable, and no severe adverse events occurred during the trial. The changes in cystatin C levels in one obese patient require some consideration. After the study termination, the cystatin C declined to pre-treatment levels. Hence, it seems highly unlikely that the patient had an affected kidney function. This

**Table 4. Adverse events at the end of Phase 1 and 2 respectively, by MedDRA SOC.**

| MedDRA SOC | MedDRA PT | Normal weight (n = 10) Week 3 No. of adverse events (No. of participants) | Obese (n = 10) Week 3 No. of adverse events (No. of participants) | Obese (n = 9) Week 12 No. of adverse events (No. of participants) |
|---|---|---|---|---|
| Participants with at least one adverse event, n | | 22 (9) | 12 (7) | 10 (7) |
| Gastrointestinal disorders | | 1 (1) | | |
| | Dyspepsia | 1 (1) | | |
| | Abdominal discomfort | | | 1 (1) |
| General disorders | | 8 (5) | 2 (2) | |
| | Exercise tolerance decreased | 1 (1) | | |
| | Fatigue | 3 (2) | 1 (1) | |
| | Hangover | 3 (3) | | |
| | Pyrexia | 1 (1) | 1 (1) | 1 (1) |
| Infections and infestations | | 6 (5) | | |
| | Influenza | 2 (2) | | |
| | Nasopharyngitis | 4 (4) | | 3 (3) |
| | Tonsillitis | | | 1 (1) |
| Injury, poisoning and procedural complications | | 2 (2) | | |
| | Arthropod bite | 1 (1) | | |
| | Limb injury | 1 (1) | | |
| Musculoskeletal and connective tissue disorders | | 1 (1) | 3 (2) | |
| | Arthralgia | | 3 (2) | |
| | Myalgia | 1 (1) | | |
| Nervous system disorders | | 3 (2) | 3 (3) | |
| | Headache | 3 (2) | 3 (3) | 2 (2) |
| Renal and urinary disorders | | | 2 (1) | |
| | Dysuria | | 2 (1) | |
| | Pollakiuria | | | 1 (1) |
| Respiratory, thoracic and mediastinal disorders | | 1 (1) | | |
| | Oropharyngeal pain | 1 (1) | | 1 (1) |
| Skin and subcutaneous tissue disorders | | | 2 (1) | |
| | Acne | | 2 (1) | |

assumption is based on three facts: Creatinine, measured at the same time as cystatin C decreased during the trial which does not indicate an affected kidney function. Furthermore, iohexol clearance test performed after the end of the study showed similar GFR levels as was obtained from creatinine calculations. Finally, mice studies with much higher doses of porous silica have not revealed any alteration of kidney function [13]. Silicic acid reverted to silicon was detected in urine, however with high individual variability. It is expected from published studies on silicon dioxide that silica is not accumulated in any tissue and excreted mainly via the kidneys [13]. This, as well as the inconclusive finding of impaired kidney function (related or not to the porous silica) in one participant, supports monitoring of kidney function as well as excluding participants with abnormal kidney values in further trials.

The laboratory data confirmed that the participants with obesity were metabolically healthy, apart from their overweight. Among the participants with obesity, serum zinc levels showed on average a statistically significant reduction. However, the changes observed are within normal range with no reason for safety concerns. Erythrocytes and hemoglobin levels decreased during Phase 1, but returned to baseline levels during Phase 2. The reasons for these changes are unknown but might be an adaptive effect or perhaps that the decrease was not due to the silica intake. Some individuals had low vitamin D levels at baseline, which further decreased during the trial. The finding that vitamin D levels decreased was not surprising, since a large decline to sun exposure occurs during fall season in Sweden [14]. The changes in retinol were inconclusive, since some participants showed higher levels and some lower levels. However, all participants stayed within the normal range. Hence, the changes in fat-soluble vitamins do not show any signal for safety concerns. The adherence for those who completed the trial was good, with approximately 1% missed doses. A possible reason for the surprisingly good adherence could be the engagement of the study staff and frequent contact between study participants and staff. In general, the reported adverse events did not cause any concerns and there were no changes observed on GI tract functions.

## Conclusion

Data from this relatively small safety study should interpreted with care and monitoring the levels of both vitamins and trace elements as well as kidney function should be considered in further trials. However, even if some biomarkers changed during this trial, these changes were of no or minor clinical relevance and adverse events observed were mild, transient and did not result in discontinuation, dose reduction or safety concern. Therefore, we conclude, in line with public data on food grade silica, that also engineered synthetic porous silica is safe to consume in relatively high doses in male humans. This opens up avenues for further studies and usage of purposely engineered porous silica in man.

## Supporting information

**S1 Checklist. TREND statement checklist.**
(PDF)

**S1 Protocol. Research protocol.**
(PDF)

**S1 Data.**
(XLSX)

## Acknowledgments

The authors like to express their gratitude to all study participants in the study. Further, we like to thank research nurse Karin Nordin for excellent collaboration and professor Claude Marcus for valuable input when designing the study.

## Author Contributions

**Conceptualization:** Emilia Hagman, Tore Bengtsson, Pernilla Danielsson.

**Data curation:** Emilia Hagman, Pernilla Danielsson.

**Formal analysis:** Emilia Hagman, Pernilla Danielsson.

**Funding acquisition:** Tore Bengtsson.

**Investigation:** Emilia Hagman, Amira Elimam, Natalia Kupferschmidt, Kerstin Ekbom, Muhammad Naeem Iqbal, Pernilla Danielsson.

**Methodology:** Natalia Kupferschmidt, Tore Bengtsson.

**Project administration:** Pernilla Danielsson.

**Resources:** Natalia Kupferschmidt, Muhammad Naeem Iqbal, Eric Johnston, Maria Lindgren, Tore Bengtsson.

**Supervision:** Emilia Hagman, Tore Bengtsson, Pernilla Danielsson.

**Validation:** Pernilla Danielsson.

**Visualization:** Muhammad Naeem Iqbal.

**Writing – original draft:** Emilia Hagman, Pernilla Danielsson.

**Writing – review & editing:** Amira Elimam, Natalia Kupferschmidt, Kerstin Ekbom, Stephan Rössner, Muhammad Naeem Iqbal, Eric Johnston, Maria Lindgren, Tore Bengtsson.

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
