## [Decision Letter · Decision Letter 0]

30 Mar 2020

PONE-D-19-24149

Oral Intake of Mesoporous Silica is Safe and Well Tolerated in Male Humans

PLOS ONE

Dear Dr Danielsson,

Thank you for submitting your manuscript to PLOS ONE. After careful consideration, we feel that it has merit but does not fully meet PLOS ONE’s publication criteria as it currently stands. Therefore, we invite you to submit a revised version of the manuscript that addresses the points raised during the review process.

The manuscript has been assessed by two reviewers; their comments are available below.

The reviewers provide overall positive evaluations but they have requested some additional and clarifications. Could you please revise the manuscript to carefully address the concerns raised by the reviewers?

We would appreciate receiving your revised manuscript by May 12 2020 11:59PM. Please include the following items when submitting your revised manuscript:

We look forward to receiving your revised manuscript.

Kind regards,

Iratxe Puebla

Deputy Editor-in-Chief, PLOS ONE

Journal Requirements:

2. Thank you for submitting your clinical trial to PLOS ONE and for providing the name of the registry and the registration number. The information in the registry entry suggests that your trial was registered after patient recruitment began. PLOS ONE strongly encourages authors to register all trials before recruiting the first participant in a study.

1) your reasons for your delay in registering this study (after enrolment of participants started);

2) confirmation that all related trials are registered by stating: “The authors confirm that all ongoing and related trials for this drug/intervention are registered”.

Please also ensure you report the date at which the ethics committee approved the study as well as the complete date range for patient recruitment and follow-up in the Methods section of your manuscript.

3. Thank you for providing the following Funding Statement: 

"This study was supported by Sigrid Therapeutics AB https://www.sigridthx.com/. The company provided the compound studied and sponsored Division of pediatrics, CLINTEC, Karolinska Institutet to conduct the study and external companies for analyzing urine. The advisory board of Sigrid Therapeutics AB was partly involved in the study design but were not involved in data collection."

We note that one or more of the authors is affiliated with the funding organization, indicating the funder may have had some role in the design, data collection, analysis or preparation of your manuscript for publication; in other words, the funder played an indirect role through the participation of the co-authors.

If the funding organization did not play a role in the study design, data collection and analysis, decision to publish, or preparation of the manuscript and only provided financial support in the form of authors' salaries and/or research materials, please review your statements relating to the author contributions, and ensure you have specifically and accurately indicated the role(s) that these authors had in your study in the Author Contributions section of the online submission form. Please make any necessary amendments directly within this section of the online submission form.  Please also update your Funding Statement to include the following statement: “The funder provided support in the form of salaries for authors [insert relevant initials], but did not have any additional role in the study design, data collection and analysis, decision to publish, or preparation of the manuscript. The specific roles of these authors are articulated in the ‘author contributions’ section.”

If the funding organization did have an additional role, please state and explain that role within your Funding Statement.

Please also provide an updated Competing Interests Statement declaring this commercial affiliation along with any other relevant declarations relating to employment, consultancy, patents, products in development, or marketed products, etc. 

Reviewers' comments:

Reviewer's Responses to Questions

**Comments to the Author**

1. Is the manuscript technically sound, and do the data support the conclusions?

Reviewer #1: Yes

Reviewer #2: Yes

2. Has the statistical analysis been performed appropriately and rigorously? 

Reviewer #1: Yes

Reviewer #2: Yes

3. Have the authors made all data underlying the findings in their manuscript fully available?

Reviewer #1: No

Reviewer #2: Yes

4. Is the manuscript presented in an intelligible fashion and written in standard English?

Reviewer #1: Yes

Reviewer #2: Yes

5. Review Comments to the Author

Reviewer #1: *** General comments: ***

In general, the manuscript appears well written, and describes a very straightforward safety study which produced very few if any warning signals.

The authors summarize anthropometrical measures and biomarkers using the mean, minimum, and maximum. Please include the standard deviation with the mean.

Although it is quite feasible to use an analysis of covariance approach to combine data from the two arms, this may be over-complicated and probably will not add much. If this were pursued, it is probably best to test for interaction effects first in each case. Again, though, it is not clear that this would add much to the presentation.

Although one cannot with certainty assign any change in values to the treatment without a control group, this is most likely not an issue for a safety study. That is because even if unrelated changes occurred(e.g., due to some environmental factor), they would be assigned as having been caused by the study agent. Therefore the study is conservative in that it may assign unrelated problems to the study agent.

A similar consideration applies to the multiple tests performed. These are uncorrected, so that the chance of missing some important signal is minimized. The trade-off is that false positive signals could get assigned to the study agent. Therefore, again, the study is conservative for safety assessments.

How specific are these results to the exact physical and chemical structure of the study agent likely to be? That is, how generalizable are these results to "all silica agents"?

*** Minor additional comments: ***

Line 127: Change to "identical looking".

Line 193: Change to "Paired t-tests were used to assess differences in anthropometrical measures and biomarkers from baseline to follow-up visits."

Reviewer #2: Mesoporous silica has emerged as a highly versatile biomaterial in academic studies, but to date very limited data is available using human test specimen. This study is the first of its kind aiming at investigating the oral tolerability of mesoporous silica microparticles. The study is sound and should be published, but need some minor revisions. The "precisely engineered porous silica" needs clarification. Does this simply just refer to the used mesoporous silica having relatively narrow pore sizes or has the material been engineered/optimized for a ceratin biological function? "Pore density" should read porosity.

6. PLOS authors have the option to publish the peer review history of their article (what does this mean?). If published, this will include your full peer review and any attached files.

Reviewer #1: No

Reviewer #2: No

---

## [Author Response · Author response to Decision Letter 0]

15 Apr 2020

Rebuttal letter

PONE-D-19-24149

Oral Intake of Mesoporous Silica is Safe and Well Tolerated in Male Humans

First, we would like to thank You for the opportunity to revise this manuscript. Please find our clarifications and answers to the academic editor and referees below. 

Journal Requirements:

Author response: Of some reason, the links above are unfortunately not working. However, we have checked that the manuscript adheres to the style requirements described at https://journals.plos.org/plosone/s/submission-guidelines#loc-author-contributions. If there still are concerns, we are happy to adjust accordingly. 

2. Thank you for submitting your clinical trial to PLOS ONE and for providing the name of the registry and the registration number. The information in the registry entry suggests that your trial was registered after patient recruitment began. PLOS ONE strongly encourages authors to register all trials before recruiting the first participant in a study.

1) your reasons for your delay in registering this study (after enrolment of participants started);

2) confirmation that all related trials are registered by stating: “The authors confirm that all ongoing and related trials for this drug/intervention are registered”.

Please also ensure you report the date at which the ethics committee approved the study as well as the complete date range for patient recruitment and follow-up in the Methods section of your manuscript.

Author response: Thank you for these suggestions. The date of ethical approval has now been added and the text says, “The study was approved by the Regional Ethical Review board, Stockholm, Sweden in May 21, 2015, file no. 2015/593-31.”, page 5, line 103.

The recommendations regarding whether phase 1 studies should be registered or not is not uniform, why the registration of the current trial was done after study initiation. Initially, we chose to comply with the European regulations (https://www.eortc.be/services/doc/clinical-eu-directive-04-april-01.pdf). This has now been added to the manuscript, page 5-6, line 106-112. All authors cannot guarantee that the company Sigrid Therapeutics AB have or will register ongoing or future trials. However, the company have done that, and this is now added to the text page 5-6, line 106-112 “Further, Sigrid Therapeutics AB confirms that all ongoing and related trials for this compound are registered.”

3. Thank you for providing the following Funding Statement: 

"This study was supported by Sigrid Therapeutics AB https://www.sigridthx.com/. The company provided the compound studied and sponsored Division of pediatrics, CLINTEC, Karolinska Institutet to conduct the study and external companies for analyzing urine. The advisory board of Sigrid Therapeutics AB was partly involved in the study design but were not involved in data collection."

We note that one or more of the authors is affiliated with the funding organization, indicating the funder may have had some role in the design, data collection, analysis or preparation of your manuscript for publication; in other words, the funder played an indirect role through the participation of the co-authors.

If the funding organization did not play a role in the study design, data collection and analysis, decision to publish, or preparation of the manuscript and only provided financial support in the form of authors' salaries and/or research materials, please review your statements relating to the author contributions, and ensure you have specifically and accurately indicated the role(s) that these authors had in your study in the Author Contributions section of the online submission form. Please make any necessary amendments directly within this section of the online submission form. Please also update your Funding Statement to include the following statement: “The funder provided support in the form of salaries for authors [insert relevant initials], but did not have any additional role in the study design, data collection and analysis, decision to publish, or preparation of the manuscript. The specific roles of these authors are articulated in the ‘author contributions’ section.”

If the funding organization did have an additional role, please state and explain that role within your Funding Statement.

Please also provide an updated Competing Interests Statement declaring this commercial affiliation along with any other relevant declarations relating to employment, consultancy, patents, products in development, or marketed products, etc. 

Author response:

- The author contribution list has now been updated as follow:

Emilia Hagman: Writing – Investigation, Formal Analysis, Original Draft Preparation

Amira Elimam: Writing – Investigation, review & editing the manuscript

Natalia Kupferschmidt: Resources, Investigation, review & editing the manuscript

Kerstin Ekbom: Investigation, review & editing the manuscript 

Stephan Rössner: Writing – Review & editing the manuscript

Muhammad Naeem Iqbal: Resources, Investigation, Visualization, review & editing the manuscript

Eric Johnston: Resources, review & editing the manuscript

Maria Lindgren: Resources, review & editing the manuscript

Tore Bengtsson: Conceptualization, Resources, Funding acquisition, review & editing the manuscript

Pernilla Danielsson: Conceptualization, Project Administration, Investigation, Formal Analysis, Data Curation, Supervision, Original Draft Preparation

- Our financial disclosure was updated as follow: 

“This study was supported by Sigrid Therapeutics AB https://www.sigridthx.com/. The company provided the compound studied and sponsored Division of pediatrics, CLINTEC, Karolinska Institutet to conduct the study and external companies for analyzing urine. Sigrid Therapeutics AB was partly involved in the study design and manuscript preparation but were not involved in data collection, data analysis or decision to publish. Sigrid Therapeutics AB also provided some support in the form of salaries or other form of remuneration at the time of the study for authors NK, SR, EJ, ML, MNI and TB. The specific roles of these authors are articulated in the ‘author contributions’ section.”

- We have also updated our Competing Interests Statement as follow: 

“The following authors have the following competing interests: NK, SR, EJ, ML, MNI and TB are or have been connected to Sigrid Therapeutics AB (employed, consultant or advisory board). This commercial affiliation does not alter our adherence to PLOS ONE policies on sharing data material. EH, AE, KE, and PD declared that no competing interests exist.”

Author response: Supporting Information captions were added at the end of the manuscript. 

Reviewers' comments:

Reviewer #1: *** General comments: ***

In general, the manuscript appears well written, and describes a very straightforward safety study which produced very few if any warning signals.

The authors summarize anthropometrical measures and biomarkers using the mean, minimum, and maximum. Please include the standard deviation with the mean.

Author response: Thank you for this suggestion. Standard deviations have now been added to Table 2.

Although it is quite feasible to use an analysis of covariance approach to combine data from the two arms, this may be over-complicated and probably will not add much. If this were pursued, it is probably best to test for interaction effects first in each case. Again, though, it is not clear that this would add much to the presentation. Although one cannot with certainty assign any change in values to the treatment without a control group, this is most likely not an issue for a safety study. That is because even if unrelated changes occurred (e.g., due to some environmental factor), they would be assigned as having been caused by the study agent. Therefore the study is conservative in that it may assign unrelated problems to the study agent.

A similar consideration applies to the multiple tests performed. These are uncorrected, so that the chance of missing some important signal is minimized. The trade-off is that false positive signals could get assigned to the study agent. Therefore, again, the study is conservative for safety assessments.

How specific are these results to the exact physical and chemical structure of the study agent likely to be? That is, how generalizable are these results to "all silica agents"?

Author response: The aim of this study was not to generalise results to all silica agents. The safety profile of synthetic amorphous silica (SAS) has already been extensively studied. For example, a recent re-evaluation by The European Food Safety Agency (EFSA) has concluded that there was no indication for toxicity of silicon dioxide (E 551) at the reported uses and use levels (EFSA J. 16(1):e05088 (2018)). However, the porous silica used in this study differed to some extent from the SAS forms evaluated in this report due to its precisely engineered pore size. The aim of this trial was to evaluate if its safety profile was still the same than other types of SAS (E551) or if the narrow pore size has an effect on its safety. 

Change to manuscript, line 75-80: “However, precisely engineered porous silica differs from other food grade silica (E551) evaluated to date by its narrower pore size distribution and it has been described to induce weight loss and lower fat content in mice (6). Therefore, it is possible that the precise pore size has an effect on the safety profile of this engineered material and in particular, this material might affect gastrointestinal functions.” 

*** Minor additional comments: ***

Line 127: Change to "identical looking".

Line 193: Change to "Paired t-tests were used to assess differences in anthropometrical measures and biomarkers from baseline to follow-up visits."

Author response: Both changes are made in the revised manuscript.

Reviewer #2: Mesoporous silica has emerged as a highly versatile biomaterial in academic studies, but to date very limited data is available using human test specimen. This study is the first of its kind aiming at investigating the oral tolerability of mesoporous silica microparticles. The study is sound and should be published, but need some minor revisions. The "precisely engineered porous silica" needs clarification. Does this simply just refer to the used mesoporous silica having relatively narrow pore sizes or has the material been engineered/optimized for a ceratin biological function

Author response: In this manuscript, precisely engineered porous silica refers to mesoporous silica with a narrow pore size distribution, as opposite to commercially available forms of synthetic amorphous silica, and in particular food grade silica (E551), that usually have a broader pore size distribution. The definition was clarified in the manuscript.

Change to manuscript, line 51-54: “Mesoporous silica compounds with narrow pore size distributions have been shown to induce weight loss in mice (6). Unlike commercial food grade silica, these compounds have narrow pore size distributions, and are therefore referred to as precisely engineered porous silica.”

 "Pore density" should read porosity.

Author response: This has been changed in the revised manuscript.

---

## [Decision Letter · Decision Letter 1]

14 Aug 2020

PONE-D-19-24149R1

Oral Intake of Mesoporous Silica is Safe and Well Tolerated in Male Humans

PLOS ONE

Dear Dr. Danielsson,

Thank you for submitting your manuscript to PLOS ONE. After careful consideration, we feel that it has merit but does not fully meet PLOS ONE’s publication criteria as it currently stands. Therefore, we invite you to submit a revised version of the manuscript that addresses the points raised during the review process.

The manuscript has been assessed by two reviewers, and their comments are appended below. Reviewer #3 asked for some modifications, please address these comments. 

We look forward to receiving your revised manuscript.

Kind regards,

Carmen Melatti

Academic Editor

PLOS ONE

Reviewers' comments:

Reviewer's Responses to Questions

**Comments to the Author**

1. If the authors have adequately addressed your comments raised in a previous round of review and you feel that this manuscript is now acceptable for publication, you may indicate that here to bypass the “Comments to the Author” section, enter your conflict of interest statement in the “Confidential to Editor” section, and submit your "Accept" recommendation.

Reviewer #2: All comments have been addressed

Reviewer #3: (No Response)

2. Is the manuscript technically sound, and do the data support the conclusions?

Reviewer #2: (No Response)

Reviewer #3: Yes

3. Has the statistical analysis been performed appropriately and rigorously? 

Reviewer #2: Yes

Reviewer #3: Yes

4. Have the authors made all data underlying the findings in their manuscript fully available?

Reviewer #2: Yes

Reviewer #3: Yes

5. Is the manuscript presented in an intelligible fashion and written in standard English?

Reviewer #2: Yes

Reviewer #3: Yes

6. Review Comments to the Author

Reviewer #2: (No Response)

Reviewer #3: This revised manuscript describes in detail a clinical study in males testing the oral intake of mesoporous silica for medical purposes. The study is well outlined and the test material thoroughly characterized. The conclusions of the study are fully supported by the results.

My only minor criticism is the use of the term "significant" at the beginning of the abstract (line 27). Significant should be reserved for the description of statistical analyses and actually there are some statistically significant effects. I would propose to delete "without significant side effects" in lines 26-27.

7. PLOS authors have the option to publish the peer review history of their article (what does this mean?). If published, this will include your full peer review and any attached files.

Reviewer #2: No

Reviewer #3: No

---

## [Author Response · Author response to Decision Letter 1]

14 Aug 2020

Dear Carmen Melatti, Academic Editor

Thank you for the reply and the opportunity to revise this manuscript. Please find our answers to the referee below. 

Sincerely, 

Pernilla Danielsson, RN, PhD

Corresponding author

Query 1-5: No comments.

Query 6. Review Comments to the Author

Reviewer #2: (No Response)

Reviewer #3: This revised manuscript describes in detail a clinical study in males testing the oral intake of mesoporous silica for medical purposes. The study is well outlined and the test material thoroughly characterized. The conclusions of the study are fully supported by the results.

My only minor criticism is the use of the term "significant" at the beginning of the abstract (line 27). Significant should be reserved for the description of statistical analyses and actually there are some statistically significant effects. I would propose to delete "without significant side effects" in lines 26-27.

Answer: Thank you for this suggestion in order to remove any confusion. The words “without significant side effects” is now removed.

---

## [Editor Report · Decision Letter 2]

18 Sep 2020

Oral Intake of Mesoporous Silica is Safe and Well Tolerated in Male Humans

PONE-D-19-24149R2

Dear Dr. Danielsson,

We’re pleased to inform you that your manuscript has been judged scientifically suitable for publication and will be formally accepted for publication once it meets all outstanding technical requirements.

Kind regards,

Hanspeter Naegeli

Guest Editor

PLOS ONE
---

## [Editor Report · Acceptance letter]

24 Sep 2020

PONE-D-19-24149R2 

Oral Intake of Mesoporous Silica is Safe and Well Tolerated in Male Humans 

Dear Dr. Danielsson:

I'm pleased to inform you that your manuscript has been deemed suitable for publication in PLOS ONE. Congratulations! Your manuscript is now with our production department. 

Kind regards, 

on behalf of

Professor Hanspeter Nägeli 

Guest Editor

PLOS ONE